# Tumor–Stromal Interactions in a Co-Culture Model of Human Pancreatic Adenocarcinoma Cells and Fibroblasts and Their Connection with Tumor Spread

**DOI:** 10.3390/biomedicines9040364

**Published:** 2021-03-31

**Authors:** Elena Prieto-García, C. Vanesa Díaz-García, Alba Agudo-López, Virginia Pardo-Marqués, Inés García-Consuegra, Sara Asensio-Peña, Marina Alonso-Riaño, Carlos Pérez, Carlos Gómez, Jorge Adeva, Luis Paz-Ares, José A. López-Martín, M. Teresa Agulló-Ortuño

**Affiliations:** 1Laboratory of Clinical and Translational Oncology, Instituto de Investigación Hospital 12 de Octubre (i+12), Av. de Córdoba S/N, 28041 Madrid, Spain; elenaprieto0102@gmail.com (E.P.-G.); cvanesa.diaz@gmail.com (C.V.D.-G.); alba.agudo@gmail.com (A.A.-L.); vpardo.imas12@h12o.es (V.P.-M.); usagi2099@gmail.com (C.P.); lpazares@seom.org (L.P.-A.); jalopezmartin@gmail.com (J.A.L.-M.); 2Proteomic Unit, Instituto de Investigación Hospital 12 de Octubre (i+12), Av. de Córdoba S/N, 28041 Madrid, Spain; inesgcg@hotmail.com (I.G.-C.); saraasensio@gmail.com (S.A.-P.); 3Biomedical Research Networking Center (CIBERER), U723, Instituto de Salud Carlos III. Av. de Córdoba S/N, 28041 Madrid, Spain; 4Laboratory of Rare Diseases, Mitochondrial &Neuromuscular Diseases, Instituto de Investigación Hospital 12 de Octubre (i+12), Av. de Córdoba S/N, 28041 Madrid, Spain; 5Pathology Department, Hospital Universitario 12 de Octubre, Av. de Córdoba S/N, 28041 Madrid, Spain; marina.alonso@salud.madrid.org; 6Medical Oncology Department, Hospital Universitario 12 de Octubre, Av. de Córdoba S/N, 28041 Madrid, Spain; cgomezm@seom.org (C.G.); drcofi@hotmail.com (J.A.); 7Biomedical Research Networking Center (CIBERONC), Instituto de Salud Carlos III, Av. de Córdoba S/N, 28041 Madrid, Spain; 8Medicine Department, Facultad de Medicina y Cirugía (UCM), Av. de Séneca, 2, 28040 Madrid, Spain; 9Department of Nursing, Physiotherapy and Occupational Therapy, Facultad de Fisioterapia y Enfermería, (UCLM), Av. de Carlos III, S/N, 45071 Toledo, Spain

**Keywords:** pancreatic ductal adenocarcinoma, cancer-associated fibroblasts, cocultures, cytokines, extracellular vesicles

## Abstract

One key feature of pancreatic ductal adenocarcinoma (PDAC) is a dense desmoplastic reaction that has been recognized as playing important roles in metastasis and therapeutic resistance. We aim to study tumor–stromal interactions in an in vitro coculture model between human PDAC cells (Capan-1 or PL-45) and fibroblasts (LC5). Confocal immunofluorescence, Enzyme-Linked Immunosorbent Assay (ELISA), and Western blotting were used to evaluate the expressions of activation markers; cytokines arrays were performed to identify secretome profiles associated with migratory and invasive properties of tumor cells; extracellular vesicle production was examined by ELISA and transmission electron microscopy. Coculture conditions increased FGF-7 secretion and α-SMA expression, characterized by fibroblast activation and decreased epithelial marker E-cadherin in tumor cells. Interestingly, tumor cells and fibroblasts migrate together, with tumor cells in forming a center surrounded by fibroblasts, maximizing the contact between cells. We show a different mechanism for tumor spread through a cooperative migration between tumor cells and activated fibroblasts. Furthermore, IL-6 levels change significantly in coculture conditions, and this could affect the invasive and migratory capacities of cells. Targeting the interaction between tumor cells and the tumor microenvironment might represent a novel therapeutic approach to advanced PDAC.

## 1. Introduction

Pancreatic ductal adenocarcinomas (PDACs) are among the most lethal tumors worldwide, with a 5-year overall survival of around 9% [1]. Unfortunately, many chemotherapeutic agents as well as radiation and targeted therapies have failed to improve survival. This lack of success has been mainly associated with late diagnosis and chemoresistance of the disease [2,3]. The increasing knowledge of cellular and molecular mechanisms involved in pancreatic carcinogenesis has also failed to translate into new therapeutic approaches. This may be due in part to the fact that most studies mainly focus on targeting tumor cells and obviate the complex surrounding microenvironment, which plays an important role in the evolution of the disease [4,5].

One of the hallmarks of PDAC is the formation of abundant and dense stroma wrapping tumor cells, termed desmoplastic reaction, which leads to an increase in the interstitial pressure, and hence to poor tissue perfusion and hypoxia. This tumor-associated stroma may represent up to 80% of the tumor mass in the majority of patients [6]. Although it was initially thought that this component was a barrier limiting tumor progression, today its protumor properties have been widely recognized [5,6,7,8,9,10]. The surrounding stroma or tumor microenvironment is composed of an altered extracellular matrix and a wide variety of cell types, including resident epithelial cells, fibroblasts, smooth muscle cells, immune and inflammatory cells, and lymphatic and vascular endothelial cells [11,12]. A population of surrounding fibroblasts known as cancer-associated fibroblasts (CAFs) play a critical role in tumor progression by stimulating survival and proliferation signaling pathways through heterotypic signaling with cancer cells [13,14].

Therefore, PDAC is formed by neoplastic cancer cells and stromal cells, and the development of the tumor requires reciprocal communication and interactions among all cell types, which cause alterations in each other by mechanisms that remain poorly understood [15]. Moreover, during tumor progression, the microenvironment progressively changes to a cancer-associated state that promotes tumor growth and metastasis. On the other hand, pancreatic cancer is a disease characterized by a highly oncogenic genetic profile [16,17,18]. Consequently, tumors are heterogeneous tissues composed of different cells with distinct characteristics, and individual tumor cells of the same tumor have different tumorigenic potentials [11,19].

Biological information between cells can also be transferred through small extracellular vesicles (EVs; exosomes, microparticles, oncosomes, etc.) released into interstitial spaces, which can be considered a snapshot of the cell’s inner contents. They are being intensively studied as promising biomarkers for treatment selection and monitoring for several cancers [20,21,22].

In this study, to elucidate the role of tumor–stromal interactions in modulating pancreatic cancer progression, the effects on migration and invasion properties of different coculture conditions between human PDAC cell lines (Capan-1 and PL45) and fibroblasts (LC5) have been studied. We show a different mechanism for tumor spread through a cooperative migration between tumor cells and activated fibroblasts. In addition, the invasive and migratory capacities of cells could be affected by IL-6 levels in the environment. Thus, our work provides further evidence that targeting the interaction between tumor cells and tumor microenvironment might be an important therapeutic approach.

## 2. Materials and Methods

### 2.1. Cell Culture and Chemical Reagents

PL-45 and Capan-1 human pancreatic cancer cell lines were obtained from LGC-ATCC (Barcelona, Spain), and an LC5 embryonic lung fibroblast cell line was kindly provided by Dr. G. Criado (Instituto de Investigación Hospital 12 de Octubre, Madrid, Spain). Differential characteristics between tumor cell lines included in the study are shown in Appendix A. Cell lines were grown according to ATCC instructions at 37 °C in humidified atmosphere with 5% CO_2_. IMDM and RPMI cell media, fetal bovine serum (FBS), penicillin, streptomycin and L-glutamine were obtained from Lonza Ltd. (Verviers, Belgium).

### 2.2. Establishment of LC5 Cell Line Constitutively Expressing EGFP

Lentivirus was generated by cotransfection of pRRL.SIN-EF1a-PGK-EGFP (encoding green fluorescent protein; #12252) with pRSV-Rev (#12253), pMD2.VSVg (#12259), and pMDL.gag/pol.RRE (#12251) plasmids in 293T cells. After 48 h, viral supernatants were collected, filtered and frozen at −80 °C until use. The LC5 cell line was infected for 24 h with 4 mL of virus-containing medium in presence of 10 µg/mL polybrene. Selection and isolation of fibroblast-GFP^+^ were performed with Sony Synergy Cell Sorter separation device (Sony Biotechnology Inc., San José, CA, USA) and based on GFP expression and incorporation of Propidium Iodide (PI) for the exclusion of dead cells. All required plasmids were from nonprofit plasmid repository Addgene (http://www.addgene.org/, accessed on 8 January 2018).

### 2.3. Cocultures of Fibroblasts and Pancreas Cancer Cell Lines

For direct cocultures between pancreatic tumor cell lines and fibroblasts, cells were cultured in a ratio of 1:1 in RPMI medium supplemented with 10% FBS. Tumor pancreatic cells were seeded 72 h before fibroblasts were added to the culture. The optimum time for these direct cocultures was experimentally established in 72 h to obtain a final ratio of tumor cells:fibroblasts of 20:80, reflecting what is described in vivo. For functional analysis, cells were separated by cell sorting based of GFP expression and incorporation of PI.

Conditioned media (CM) were prepared from fibroblasts and pancreatic cancer cell lines. Cells were grown in 75 cm flasks until semiconfluent conditions were achieved, washed three times with phosphate-buffered saline (PBS), and then incubated in 1% of FBS medium for another 48 h. Next, supernatants were collected, centrifuged at 2000 rpm for 10 min, sterile filtered (0.45 µm pore), and immediately used for assays of indirect cocultures.

### 2.4. Wound Healing, Migration, Proliferation and Invasion Assays

For the wound-healing assay, cells were seeded on 6-well plates and left to reach 80% confluence. A single wound per well was scratched with a sterile 200 µL pipette tip within the monolayers and then washed with PBS to remove detached cells. CM from tumor cells was added to fibroblasts, and similarly CM from fibroblasts was added to tumor cells, and the rate of wound closure was observed in the following 24, 48, and 72 h. Wound closure was monitored using a Nikon TE2000-S microscope.

Transwell migration assays were carried out using 8.0 µm polycarbonate cell culture inserts (Transwell^®^ Permeable Supports, Corning Inc., New York, NY, USA). Cells were serum-starved 24 h before the start of the experiments. Tumor cells were seeded at 1 × 10^−5^ per well in 2 mL growth medium in the upper chamber (cell culture inserts) of 6-well plates. The same number of fibroblasts was seeded in the lower wells. Cells were allowed to migrate for 72 h toward the underside of the membrane to the lower chamber. After that, cells attached to the upper side of the filter were removed by wiping with a cotton swab, and the migrated cells on the lower surface were fixed with formaldehyde 4% in PBS and permeabilized with absolute methanol in darkness. After several washes with PBS, cell nuclei were stained with 4’-6-diamino-2-phenylindole (DAPI).

Proliferation was examined by using WST-1 assay (Roche, Mannenheim) according to the manufacturer’s protocol. Cells were cultured at 3000 (tumor cells) or 2000 (fibroblasts) per well in 96-well flat-bottom plates and allowed to attach for 24 h, and then growth medium was replaced by CM from fibroblasts or tumor cells for another 48 h. Results are expressed as percentage of cells growing in CM relative to cells growing as monocultures under standard conditions.

We used Ibidi culture-insert 2 well in µ-dish 35 mm plates with a defined 500 µm cell free gap (Inycom Biotech, Zaragoza, Spain) for migration assays. Tumor cells were seeded on one side of the insert and fibroblast-GFP^+^ on the opposite side. After 24 h for cell adhesion, the insert was removed, and migration was allowed. This system allowed us to observe the migratory behavior of each cell type under the indirect influence of the other.

### 2.5. Western Blot Analysis

Whole cell lysates were prepared using MCL1 lysis buffer in the presence of protease and phosphatase cocktail inhibitor (Sigma-Aldrich, St Louis, MO, USA) following the manufacturer’s protocol. Protein lysates were subjected to SDS-PAGE on 10% polyacrylamide gel and transferred onto PVDF membranes. Membranes were probed for the protein levels of E-cadherin, using α-tubulin as loading control and specific primary antibodies (#4065, Cell Signaling Tech. Inc., Danvers, MA, USA, and #sc-5286, Santa Cruz Biotechnology, Inc., Dallas, TX, USA, respectively). Specific bands for each protein were visualized by a WesternBright™ kit (Advansta, San Jose, CA, USA).

### 2.6. Enzyme-Linked Immunosorbent Assay (ELISA) and Human Cytokine Array

Culture supernatant was collected from each growth condition, and keratinocyte growth factor (KGF-7) concentration was determined using the Quantikine human KGF/FGF-7 Immunoassay (R&D Systems Inc., Minneapolis, MN, USA). An ExoQuick-TC^®^ ULTRA EV Kit and ExoELISA-ULTRA CD63 Kit (System Biosciences, Palo Alto, CA, USA) were used for extracellular vesicle isolation and quantification, respectively. For the semiquantitative detection of 23 human proteins in cell culture media, the human cytokine antibody array C1 (AAH-CYT-1; RayBiotech, Norcross, GA, USA), was used (Appendix A). All determinations were performed according to the manufacturer’s instructions.

### 2.7. Confocal Immunofluorescence Microscopy

Direct and indirect cocultures of fibroblast-GFP^+^ and tumor pancreatic cells were seeded on glass coverslips. Cells were washed twice with warm PBS, fixed in 4% paraformaldehyde and processed for immunofluorescence staining. First, cells were incubated overnight with mouse anti-α-Smooth Muscle Actin (α-SMA) antibody (n1584, Dako, Carpinteria, CA, USA) at 4 °C. After thorough washing with PBS, cells were incubated for 1 h with Alexa Fluor^®^ 594-conjugated goat antimouse antibody and DAPI. Cells were mounted with Fluorsave™ Reagent (Calbiochem, San Diego, CA, USA) and examined in a Zeiss Axioplan 2 confocal microscope (Zeiss Microscopy, Göttinger, Germany).

### 2.8. Transmission Electron Microscopy (TEM)

Cells were fixed in 2.5% glutaraldehyde at pH 7.2 for 24 h, and later in 1% OsO4 in a 0.1 M cacodylate buffer for 1 h. Then, the samples were spun to obtain pellets. The pellets were fixed in 1% uranyl acetate for 30 min, then dehydrated in a series of graded ethanol steps, and finally embedded in epoxy resin. Thin sections were performed and stained with toluidine blue. Ultrathin sections were obtained from representative areas and were double stained with lead citrate and uranyl acetate and viewed under a JEOL JEM-1011 microscope (JEOL USA Inc, Peabody, MA, USA).

### 2.9. Statistical Analysis

Each experiment was performed in triplicate in at least three independent experiments. The analysis of differences was performed using a Mann–Whitney U test and variance analysis for repeated measurement data. Results were analyzed using the software GraphPad Prism version 6.0 (GraphPad software, San Diego, CA, USA) and SPSS v21.0 (SPSS Inc., Chicago, IL, USA). The value *p* < 0.05 was considered as significant.

## 3. Results

### 3.1. Cocultures of Pancreatic Tumor Cells and Fibroblasts-GFP+

First, we established a fibroblast cell line constitutively expressing GFP protein (LC5-GFP^+^), in which the percentage of fibroblast-GFP^+^ was higher than 97% (Figure 1A). Experimental design and images from direct cocultures between PDAC and LC5-GFP+ cells at 72 h are shown in Figure 1B. It is noteworthy that fibroblasts in the vicinity of the tumor cells surface established close contacts to these cells, being arranged around the tumor cells in a way that maximized the contact between cells.

With the aim of evaluating the importance of the cell–cell adhesion and juxtacrine signaling versus paracrine signaling between cancer and stromal cells in the pathophysiology of pancreatic cancer, direct and indirect cocultures with CMs were used in the following assays.

### 3.2. Coculture Modifies the Expression of Activation Markers in Tumor Cells and Fibroblasts

To assess activation of fibroblasts, LC5-GFP^+^ cells were seeded in direct cocultures with tumor cells or indirect cocultures with CM from Capan-1 or PL-45 cells. Representative pictures of α-SMA immunofluorescence staining are shown in Figure 2A. LC5-GFP^+^, Capan-1, and PL-45 cells do not express the α-SMA activation marker when they grow alone at monocultures (rows 1, 2, and 5). However, we can see that fibroblasts express high level of α-SMA both in direct and indirect cultures with tumor cells (rows 3, 4, 6, and 7). Quantification of LC5-GFP+ cells expressing α-SMA is shown in Figure 2C. However, this fibroblast activation did not result in increased tumor cell proliferation since no significant differences between indirect and monoculture conditions were found in cell proliferation assays (Figure 2D).

In addition, different FGF-7 expression levels between cells grown alone or in cocultures were found. Thus, FGF-7 levels in coculture supernatants were higher than in monoculture supernatants were (Figure 3A, *p* < 0.0001). These results show that fibroblasts under the coculture conditions described here acquired the so-called myofibroblast or CAF phenotype.

Similarly, tumor cells were also affected by the presence of fibroblasts in coculture. To assess the induction of the epithelial mesenchymal transition (EMT) process, we isolated tumor cells by sorting and quantified the protein levels of E-cadherin by immunoblotting (Figure 3B,C). E-cadherin is a cell adhesion molecule necessary for maintenance of intercellular contacts and cellular polarity in epithelial tissues, and is a key feature of EMT. We can see that in both direct and indirect cocultures, pancreatic tumor cells decrease their expression of E-cadherin, which is attributed to an EMT process.

### 3.3. Cytokines Secretion Changes under Coculture Conditions

It has been reported that cytokines and growth factors secreted by CAF or tumor cells are associated with tumor invasion and migration. Thus, the effects of direct and indirect cocultivation of fibroblasts and tumor cells on cytokine production were examined next. We collected the supernatants from monocultures, direct and indirect cocultures, and quantified the differences in protein expressions of 23 specific human cytokines by a human cytokine antibody array (Figure 3D–G; Appendix A). Relative levels of the cytokines are differentially expressed between fibroblasts; Capan-1 and PL-45 monocultures are shown in Figure 3D. Then, we compared the expressions of these proteins in direct and indirect cocultures versus respective monocultures (Figure 3E, versus fibroblasts monoculture; Figure 3F, versus Capan-1 monoculture; Figure 3G, versus PL-45 monoculture). We found that the GM-CSF growth factor, and the cytokines GRO a/b/g, GRO α, IL-6, IL-8 and RANTES were markedly increased in LC5-cancer cell cocultures compared to LC5 monoculture. Curiously, GM-CSF, GRO a/b/g and GRO α, showed a considerable decrease only when LC5 cells were grown with CM of PL-45. The high expression of IL-6 in cocultures of fibroblasts and tumor cells compared to tumor cell monocultures is noteworthy (Figure 3E–G). However, we observed an increase in chemokine RANTES and growth factor GM-CSF levels in coculture of fibroblasts with Capan-1 cells, but a decrease in coculture of fibroblasts and PL-45 cells. The cocultivation of fibroblasts and PL-45 cells was the one that showed the most differences in the pattern of cytokine secretion depending on the coculture conditions, mainly in relation to IL-8, GRO α and GRO a/b/g levels.

### 3.4. Coculture Modifies Motility of Tumor Cells and Fibroblasts

The migratory behavior of cells in direct or indirect coculture was examined by wound-healing and transwell migration assays. Figure 4A–C show representative images of Capan-1 cells grown in direct coculture with fibroblast-GFP^+^. Both cell types migrate into the wound, but curiously fibroblasts migrate predominantly as single cells, whereas tumor cells migrated as a tightly packed sheet surrounded by fibroblasts.

To explore whether cell–cell contact is necessary for migration activity, we cultured Capan-1 or PL-45 tumor cells with fibroblast CM for 3 days (Figure 4E), and vice versa—LC5 was cultured with CM from each tumor cell line (Figure 4F) and their migration activity was assayed to assess wound-healing. Only PL-45 cells exposed to LC5 supernatant showed a weak migration increase at 48 h (Figure 4E).

On the other hand, Capan-1- and PL-45-CM differed in their ability to affect the movement of LC5 cells, with Capan-1-CM being a more potent inducer (Figure 4F). It is noteworthy that fibroblasts grown with Capan-1-CM travel to the center of the wound, invading the entire field. In contrast, fibroblasts cultured with PL-45-CM do not leave the edge of the wound, suggesting that wound healing occurs rather by cell proliferation than by fibroblast migration.

Next, Capan-1 or PL-45 cells were cultured in the upper chamber of a transwell migration system for 3 days. The number of invasive tumor cells increased significantly when fibroblasts were seeded in the lower chamber compared to complete growth medium control. This is substantial in the Capan-1 cell line, but even more so in the case of the PL-45 tumor cell line (Figure 4D).

Last, we carried out migration assays in the Ibidi culture-insert plates. Tumor cells were seeded on one side of the insert and LC5-GFP+ on the other one. As a control, LC5-GFP+ cells were seeded on both sides of the insert (first column, Figure 4G). Fibroblasts in coculture with PL-45 cells exhibited a migration similar to that observed in the control (Figure 4G, columns 4 and 5). However, fibroblasts in coculture with Capan-1 cells increased their migratory capacity, leaving the edge of the culture and moving towards the tumor cells (Figure 4G, columns 2 and 3). Moreover, in the vicinity of contact between both cell types, fibroblasts were arranged so as to increase the surface area of contact with tumor cells. These results are in agreement with those obtained previously with direct and indirect cocultures.

### 3.5. Identification and Quantification of Extracellular Vesicles (EVs) in Cocultures

Since there is ample evidence that EVs play a crucial role in intercellular communication, we decided to study the secretion of these vesicles in our coculture assays. Visualization was realized by TEM (Figure 5A–E). Although TEM is not a quantitative technique and we cannot differentiate between tumor cells and fibroblasts in cocultures, we found a higher number of EVs per cell under coculture versus monoculture conditions. EVs displayed typical spherical morphologies and fell within the expected size range of exosomes (30–150 nm). We therefore isolated and quantified EVs from supernatant samples, and we could see that exosomes from cocultures had a higher amount of protein than those from monocultures (Figure 5F).

## 4. Discussion

Pancreatic tumors are heterogeneous tissues composed of tumor and nontumor cells embedded within a dense desmoplastic stroma [6,23]. This is clear evidence that stroma is not just a static mechanical barrier but is thought to be an active participant involved in tumor initiation, progression, and metastasis [24]. Thus, the study of pancreatic cancer cell behavior in monocultures does not faithfully reflect what is observed in vivo, given that all tumor components have a significant impact on the biological properties of malignancy. In this regard, in the work here exposed we used in vitro coculture models in an attempt to recapitulate some of the stromal–tumor connections to assess the impact of this partnership on migration and metastasis initiation in PDAC.

Fibroblasts are the most common component of tumor stroma, and CAFs provide a favorable microenvironment with tumor-promoting properties, stimulating the progression of primary tumors via paracrine signaling [25], and serving as a niche to support the metastatic colonization [26,27,28]. Thus, the association between cancer cells and CAFs leads to alterations of the biological properties of both types of cells through bidirectional tumor stroma crosstalk.

In our coculture model, KGF/FGF-7 secretion was upregulated; it has previously been postulated than CAFs could increase the viability and migration of tumor cells in a paracrine manner through this growth factor [29]. In addition, α-SMA is the most widely used marker to assess CAF activation status and is associated with worse clinical outcomes [30,31].

Cell migration is an important factor in the spread of pancreatic cancer and is a key step in the series of events that lead to metastasis. In the wound-healing assays of direct coculture, we can see that groups of cells move into the wound. In these clusters, fibroblasts are arranged surrounding tumor cells. In addition to this curious migration formation, it is worth noting the amoeboid characteristics of tumor cell movements. Amoeboid migration facilitates cell position changes without cell remodeling but moves with low adhesion force and high cell deformability. Instead, collective migration takes place when cells retain cell–cell adhesions and coordination among them. These migration modes are interconvertible, and tumor cells use both mechanisms in an adaptive way, exhibiting their great plasticity [32]. In our work, this peculiar way of migrating set, tumor cells and surrounding fibroblasts in the middle of a wound could indicate that both types of cells actively cooperate to migrate to other organs as a single invasion. Previous reports have shown that single cancer cells may proliferate via interactions with stromal cells derived from other organs when they form metastatic tumors [33]. However, these mechanisms are not mutually exclusive, and fibroblasts could come from either the primary tumor or metastatic sites.

Using a traditional migration assay setup, we show that only Capan-1 cells were able to induce significant CAF migration as individually moving cells. Similarly, Ishii et al. have reported that lung fibroblasts cocultured with human cancer cell line Capan-1 showed significantly higher migration activity than fibroblasts alone [34]. In addition, Karagiannis et al. showed in cocultures of colon cancer cell lines and fibroblasts that the CAF migratory behavior was context-dependent, varying from totally individual to collective configurations [35]. On the other hand, we must not forget the different origins of the tumor cells used in this study: PL-45 cells come from a primary tumor, while Capan-1 comes from a metastatic site. Given the results obtained in our study, we could speculate that cells in the metastatic site attract fibroblasts to them, while cells from the primary tumor are more stimulated by the fibroblasts to start invasion. Therefore, fibroblasts would have dissimilar migratory responses upon stimulation with different pancreatic cancer cells, such as Capan-1 or PL-45, and different microenvironments.

Activated fibroblasts can promote invasive growth by cell–cell contacts or by paracrine diffusible signals, which can also induce EMT of tumor cells [36,37]. On the other hand, we have shown an important decrease in the expression of the epithelial marker E-cadherin in tumor cells from our coculture model, suggesting that EMT occurs in this system and would facilitate the initiation of metastasis [4]. However, variable expression of E-cadherin in human pancreatic cancer samples and commercial cell lines has been confirmed in several studies, supporting the hypothesis that PDAC cells exhibit a hybrid EMT-related phenotype [38].

In the crosstalk between fibroblasts and cancer cells, two closely related pathways have been described: one pathway where stroma cells affect cancer cells response, and another where cancer cells trigger a response in the stroma cells [37,39]. Both responses could be mediated by different secretions of cytokines from the cells involved. Furthermore, we found that the levels of IL-6, in addition to GM-CSF and RANTES, changed significantly in coculture conditions with respect to monocultures; this could affect the invasive and migratory capacities of cells. Increased IL-6 levels have previously been related to protumorigenic functions of CAFs, promoting contractibility and extracellular matrix remodeling [40,41]. Curiously, we have not found a change in TGF-β levels between the different coculture conditions. IL-6 is also a common cytokine that enhances TGF-β signaling, resulting in EMT, and stimulates tumor progression. However, TGF-β signaling is multifaceted and cell type specific in PDAC [42]. Thus, in the heterogeneous cancer-associated fibroblasts population, there may be a subset of inflammatory CAFs expressing proinflammatory cytokines, which could make up the majority in our coculture models.

Exosomes have recently been described as communication tools between cells, and they could mediate interactions between tumor cells and the microenvironment [22,43]. They are secreted from almost all cell types, although tumor cells secrete more EVs compared to normal cells, playing an essential role in inducing metastasis and remodeling the primary microenvironment. Thus, it has been shown that EVs from fibroblasts promote invasive and chemoresistant behaviors in cancer cells [44,45], whereas EVs from tumor cells can reprogram fibroblasts into CAFs. In addition, it has been described that an increased amount of protein per exosomes predict disease progression [46,47]. Our results are in agreement with previously published data that indicate that cells in cocultures produce a greater number of exosomes, and these possess a higher amount of protein that cells in monoculture.

A growing body of literature covering tumor–stroma interactions points to future approaches for cancer treatments being based on both malignant and microenvironmental cells [24,48,49]. Since pancreatic cancer cells are surrounded by a thick and poorly perfused stroma, which halts penetration of otherwise effective treatments, it would be reasonable to think that incorporating a stromal depleting agent into therapeutic treatments or disrupting the crosstalk between stroma and cancer cells would sensitize chemoresistant tumors and improve the outcome of patients [50,51,52]. Similarly, exosomes could be used in antitumor therapies by exploiting their characteristic properties [53].

Finally, an important limitation of our study is that it focuses on cancer-associated fibroblasts as the leading stromal element interacting with pancreatic tumor cells. However, many other cell types in the stroma are known to influence tumor cell behavior, such as immune and inflammatory cells, adipocytes, stellate cells, and endothelial cells. On the other hand, the fibroblasts used in this work are embryonic lung cells without prior contact with primary tumors or metastatic niches, which acquired CAF conditions in cocultures with pancreatic tumor cells. Two different PDAC cell lines were used in our coculture model, and we cannot generalize the results to the entire spectrum of pancreatic cancer. In this study, we have achieved some relevant results, but also accept the incomplete approximation of the in vitro setups, as they are not always able to simulate the high tissue complexity found in vivo.

## 5. Conclusions

Collectively, we have shown a coculture model of tumor cells and fibroblasts that recapitulates some features occurring in human PDAC and might be used for a better understanding of tumor–stromal interactions and to identify new targets for pharmacological intervention. We show different migratory properties in tumor and stromal cells and a different mechanism for tumor spread through the joint migration of tumor cells and fibroblasts. Taken together, our results suggest that direct contact between fibroblasts and Capan-1 cells is involved in the progression of tumors, in addition to the paracrine-promoting effect of growth factor or chemokines such as GM-CSF or IL-6. However, CAFs could lead PL-45 migration mainly though paracrine mechanisms, in addition to juxtacrine ones. Although cancer–stroma interactions may be a promising target for new therapeutic approaches, these should focus on each tumor’s histotype due to the variability of tumor cells and the role played by CAFs in each tumor.

## Figures and Tables

**Figure 1 biomedicines-09-00364-f001:**
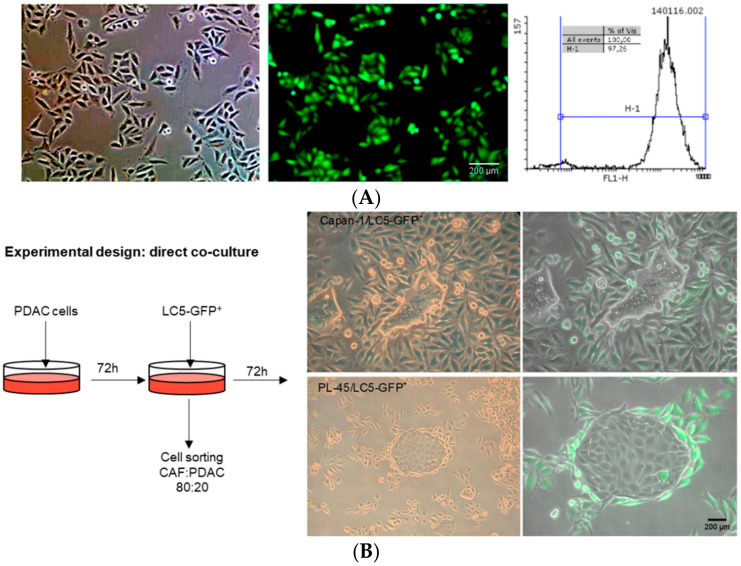
Photographs of phase contrast and fluorescence microscopy of the selection of fibroblast LC5-GFP^+^ after FACS sorting. The percentage of GFP expressing fibroblasts was higher than 97% (**A**). Strategy scheme of direct cocultures: representative images at 72 h of pancreatic tumor cells (Capan-1 or PL-45) and fibroblast LC5-GFP^+^ (**B**). Green cells are GFP-transfected fibroblasts.

**Figure 2 biomedicines-09-00364-f002:**
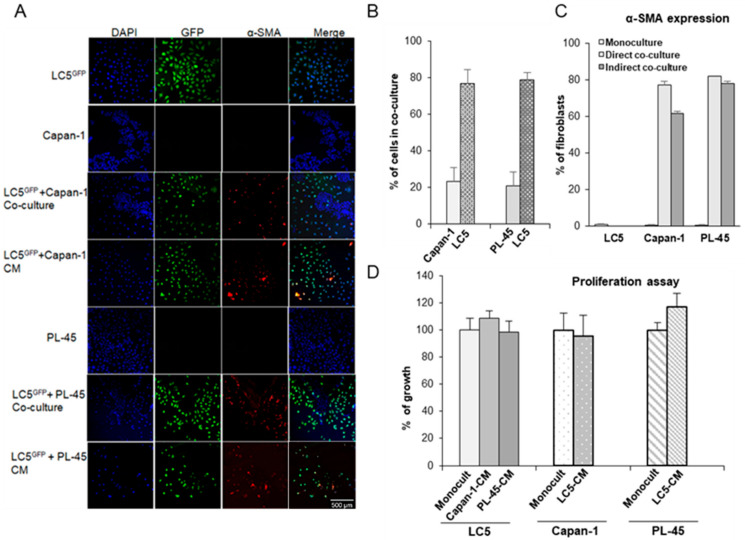
Coculture modifies the expression of activation markers in tumor cells and fibroblasts without affecting proliferation. Activated fibroblast LC5-GFP^+^ express α-SMA in direct and indirect cocultures with pancreatic tumor cells lines. Rows 1, 2, and 5: indicated cells in monoculture; rows 3 and 4: LC5^GFP^ in direct or indirect coculture, respectively, with Capan-1 pancreatic tumor cell line; rows 6 and 7: LC5^GFP^ in direct and indirect coculture with PL-45 pancreatic tumor cell line (**A**). Representative bar graph showing percentage of fibroblast and tumor cells in direct cocultures is shown. Data were obtained by counting cells in three different fields of three different photographs (**B**). Representative bar graph showing percentage of fibroblasts expressing α-SMA in direct or indirect cocultures with respect to monoculture is shown. Data were obtained from three independent experiments and are reported as mean ± SD (**C**). Proliferation assays of cells growing in conditioned medium from fibroblasts or tumor cells are shown with respect to cells growing as monocultures. Data are presented as the mean ± SD of three independent experiments (**D**). LC5^GFP^: LC5 fibroblasts expressing GFP protein. CM: Conditioned medium. The green color indicated GFP-transfected fibroblasts, the red color indicates α-SMA, and the blue color indicated the nuclei.

**Figure 3 biomedicines-09-00364-f003:**
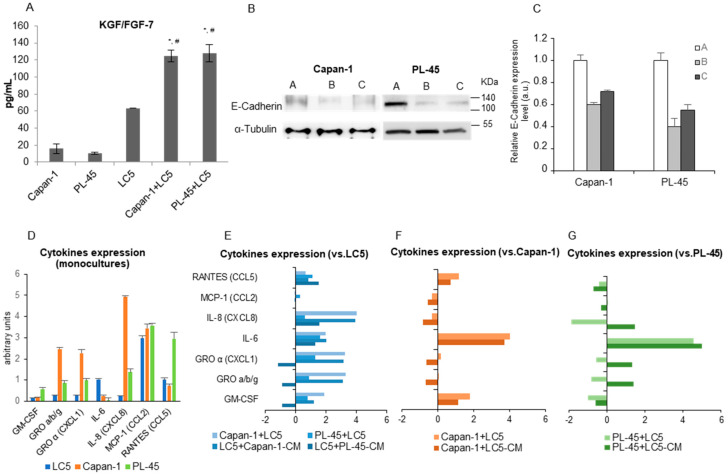
Changes in protein expression and cytokine secretion in coculture models. KGF/FGF-7 concentration from supernatants of monoculture and fibroblasts-pancreatic tumor cells cocultures. FGF-7 concentration in cocultures was significantly higher than that in monocultures (*—LC5, #—corresponding tumor cell line; *p* < 0.0001). Data represent mean ± SD of three independent assays (**A**). Expression of epithelial mesenchymal transition (EMT) marker E-cadherin in pancreatic tumor cells by immunoblotting is shown. Loading control—α-Tubulin (**B**). Representative quantification of E-cadherin expression from Western blot is shown. A: cells grown alone at monoculture. B: cells grown at cocultures with fibroblasts (protein extracts obtained after sorting), and C: cells grown with CM from fibroblast (**C**). Quantification of secretome dot blots from LC5, Capan-1 and PL-45 cells growing as monocultures according to AAH-CYT-1 human cytokine antibody array is shown—represents the molecules most significantly expressed (**D**). Relative differences are shown by cytokine expression levels compared to LC5 monoculture (**E**), compared to Capan-1 monoculture (**F**), and compared to PL-45 monoculture (**G**). *X*-axis values shown are log2 of the normalized levels.

**Figure 4 biomedicines-09-00364-f004:**
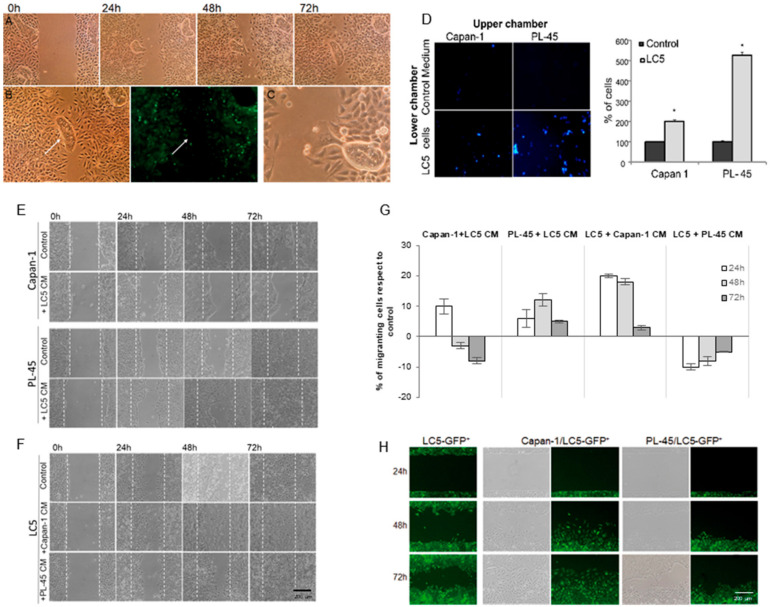
Migration and invasion assays in coculture conditions. Wound-healing assay in direct cocultures of LC5-GFP^+^ and Capan-1 tumor cell line. Photographs taken after 24, 48, and 72 h of coculture are shown (**A**). Images of cell migration in phase contrast (left) and fluorescence microscope (right) are presented. The white arrow points to a group of tumor cells migrating like a packed sheet (**B**). Detail of tumor cells invading the wound: enlarged figure depicts tumor cells that are polarized with cytoplasmic protrusions facing the wound in an amoeboid movement (**C**). Transwell migration assay: representative photographs of Capan-1 and PL-45 cells that have invaded into the 8 µm pore membrane filter and histograms of migrated tumor cells compared to control (100%) are shown. Migrated cells were quantitated by counting cells in 10 random fields in each sample. Data are presented as mean ± SD from three independent experiments (* *p*< 0.0001) (**D**). Wound scratch assay: treatment of pancreatic tumor cell lines with conditioned medium (CM) from fibroblasts. Control: tumor cells in growth medium (**E**). Treatment of LC5 fibroblasts with conditioned medium (CM) from Capan-1 or PL-45 pancreatic tumor cell lines. Control: LC5 fibroblasts in growth medium (**F**). Bar graph shows the percentage of migratory cells with respect to their controls in wound scratch assays. Migration was quantified by counting the 10 cells furthest from the edge of the wound in 5 random fields from 3 different photographs. Data are presented as mean ± SD (**G**). Wound-healing assay in 2 well Ibidi culture-insert. First column: images in fluorescence microscope of cell migration at 24, 48 and 72 h of LC5-GFP+ cells seeded on both sides of the insert. Columns 2 and 3: images of cell migration in phase contrast (column 2) and fluorescence microscopes (column 3) of Capan-1 tumor cells seeded in the upper side of the image and LC5-GFP^+^ seeded in the lower side. Columns 4 and 5: images of migration with PL-45 tumor cells in the upper side and LC5-GFP+ in the lower side (**H**). The green color indicated GFP-transfected fibroblasts. Scale bar indicates 200 µm.

**Figure 5 biomedicines-09-00364-f005:**
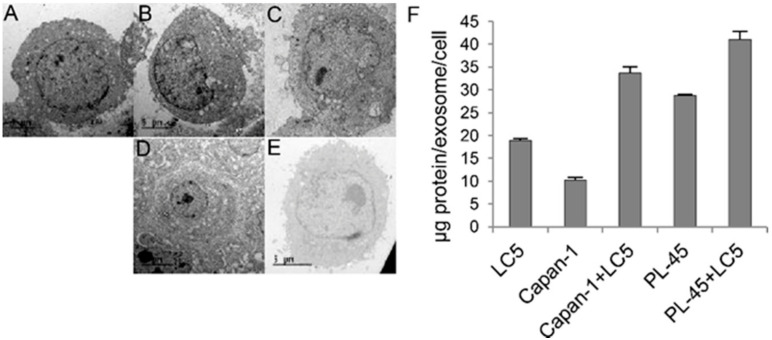
Representative images by transmission electron microscopy (TEM) of cells growing for 72 h. LC5 cells growing at monoculture (**A**). Capan-1 cells in monoculture (**B**). PL-45 cells in monoculture (**C**). Capan-1 and LC5 cells in coculture (**D**). PL-45 and LC5 cells in coculture (**E**). Quantification of protein by exosome in monoculture and coculture conditions (**F**). Scale bar indicates 5 µm.

## Data Availability

Not applicable.

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
