# Peer review of "Tumor–Stromal Interactions in a Co-Culture Model of Human Pancreatic Adenocarcinoma Cells and Fibroblasts and Their Connection with Tumor Spread"

_biomedicines, 2021, doi:10.3390/biomedicines9040364_

Round 1
Reviewer 1 Report
In the manuscript “Tumor-stromal interactions in a co-culture model of human pancreatic adenocarcinoma cells and fibroblasts, and their connection with tumor spread” Prieto-García E. and colleagues investigate the behavior and characterize the interactions between pancreatic cancer cells and fibroblast, one of the main components of tumor stroma. Understanding this cross-talk is particularly relevant for designing more effective therapy against pancreatic cancer, which is characterized by the presence of a dense stroma.
To this goal, the authors studied the effects of direct and indirect co-cultures of both pancreatic ductal adenocarcinoma (PDAC) cells and fibroblasts and their migratory effect. Furthermore, the relative presence growth factors and cytokines secreted in mono- or co-cultures were presented. Overall, the authors concisely report the influence of fibroblasts in various processes and also demonstrate increased protein estimation in extracellular vesicles derived from co-cultures.
The study confirms previous findings, e.g. the crosstalk between these two cell populations and the mutual support towards activation and migration. Interestingly the authors reported that different chemokines were increased in indirect co-cultures (IL-6).
The data are presented in a clear way, and are convincing, however several revisions are needed in order to improve the quality of the manuscript. More precisely:
- It is not specified the tissue from which the embryonic fibroblasts were isolated. Is it a tissue different from the tumor primary site or do these fibroblasts derive from a well-known metastatic site? This piece of information is relevant for the study, to discriminate among different CAFs sub-populations, and should be also mentioned in the limitations of the study.
- In line 217 ‘To check activation of fibroblasts…’, authors demonstrated via immunofluorescence staining of α-SMA the activation of these fibroblasts. Was α-SMA staining also checked (as a positive control) with other known growth factors or cytokines that are known to activate fibroblasts, such as TGF-β? If so, please add to Figure 2A or in supplementary figures.
- Lines 247-248 “We can see that in both direct and indirect co-cultures, pancreatic tumor cells decrease their expression of E-cadherin, which would be attributed to an EMT process.” These statements should be supported by at least one additional EMT marker as evidence (either gain/loss of expression of a mesenchymal/epithelial marker, e.g. vimentin).
- It would add value to the study to perform experiments in which isolated EVs are given to cells, instead of whole conditioned media, to evaluate whether they are responsible for fibroblast activation and/or increased migration.
- We appreciate authors’ declaration that findings cannot be generalized due to missing information about other cellular components taking parts in physiological situation. However, after specifying the tissue of origin of the fibroblasts the discussion part should also mention whether the results are relevant for the tumor primary site (pancreas) or for a metastatic niche.
- As the authors point out, CAPAN-1 cells migrate as a packed sheet, and this collective migration implies that cell-cell contact is maintained to a certain extent. As it is discussed, it could be an effect of partial-EMT, and more insights could be gained by looking at the expression of E-cadherin of cells in the sheets (for example by immunofluorescence).
- It would be interesting also to see if these findings would be validated by co-culturing tumor cells with pancreatic stellate cells.
- Finally, it is curious the different effect that the two tumor cell lines can have on fibroblasts migration. They are both classified as epithelial, but Capan1 are isolated from a metastatic site, while PL-45 are isolated from the primary tumor. This could be one reason of the different behavior, as they would interact with different microenvironments. It could be speculated that cells in the metastatic site attract fibroblasts to them, while cells from the primary tumor are more stimulated by the fibroblasts to start invasion. A broader panel of cell lines could offer the opportunity to find patterns to better understand the tumor-stromal interaction.
Specific points about the figures:
- Figures 2A shows images of direct co-cultures of PDAC-fibroblast cells, but the 20:80 ratio does not appear evident (more likely 50:50). Please add cell number quantification in direct co-cultures reflecting 20:80 ratio as declared in the experimental design.
- In Figure 3B E-cadherin expression was detected via Western blot where the decrease in E-cadherin expression was indicated to be attributed to an EMT process. Was vimentin expression also determined before?
- Figures 4E-G, and the relative paragraph 3.4, explaining the migratory capacity of cells are hard to understand and quantify. Please include graphs with percentages of migration for each cell line and condition to help the reader easily interpret the results.
- Description of figure 4F: it is stated that the control in this experiment were tumor cells in growth medium. Since the experiment was investigating the effect of tumor cells on invasion of LC5-GFP+, a more appropriate control would be LC5-GFP+ in growth medium.
- In Figure 5 A-E (TEM images) arrows pointing at extracellular vesicles (if visible) would help the reader understanding the figure.
Please check spelling and grammar within the article.
Author Response
1. It is not specified the tissue from which the embryonic fibroblasts were isolated. Is it a tissue different from the tumor primary site or do these fibroblasts derive from a well-known metastatic site? This piece of information is relevant for the study, to discriminate among different CAFs sub-populations, and should be also mentioned in the limitations of the study.
We thank the reviewer for pointing out this flaw. Fibroblast cell line is derived from embryonic lung, not from a primary tumor or metastatic site. Following the reviewer's instructions, we have added this information in the manuscript (Materials and Methods) and highlighted its importance in the limitations of the study (Discussion):
“On the other hand, fibroblasts used in this work are embryonic lung cells without prior contact with primary tumors or metastatic niches, which acquired CAF conditions in co-cultures with pancreatic tumor cells”.
2. In line 217 ‘To check activation of fibroblasts…’, authors demonstrated via immunofluorescence staining of α-SMA the activation of these fibroblasts. Was α-SMA staining also checked (as a positive control) with other known growth factors or cytokines that are known to activate fibroblasts, such as TGF-β? If so, please add to Figure 2A or in supplementary figures.
We agree with the reviewer that it would have been a good idea to check fibroblasts activation with other known activators. Unfortunately, we did not do it at experimentation time. However, in our modest opinion, this fact does not detract from our results, in which the activation of fibroblasts can be observed (by the expression of α-SMA) in conditions of co-culture with tumor cells.
3. Lines 247-248 “We can see that in both direct and indirect co-cultures, pancreatic tumor cells decrease their expression of E-cadherin, which would be attributed to an EMT process.” These statements should be supported by at least one additional EMT marker as evidence (either gain/loss of expression of a mesenchymal/epithelial marker, e.g. vimentin).
In order to meet this requirement of the reviewer, we performed a stripping of the membranes in which the expression of E-cadherin was verified, and we hybridized them with an anti-Vimentin antibody. Unfortunately we were unable to see the expression of this protein in any of the conditions studied in these membranes. In our opinion, this was due to the fact that the antibody did not work properly. We are aware of the importance of this result, but we would have to acquire a new antibody and perform the full experiment from the beginning again. Given the time allowed to reply to the reviewers, it has been impossible for us to do so.
4. It would add value to the study to perform experiments in which isolated EVs are given to cells, instead of whole conditioned media, to evaluate whether they are responsible for fibroblast activation and/or increased migration.
Again, we agree with the reviewer's suggestion. Isolation of EV requires the use of ultracentrifuges, but our laboratory does not have them. In order to satisfy this requirement, we would have to restart the experiment again and seek external collaboration to complete it.
5. We appreciate authors’ declaration that findings cannot be generalized due to missing information about other cellular components taking parts in physiological situation. However, after specifying the tissue of origin of the fibroblasts the discussion part should also mention whether the results are relevant for the tumor primary site (pancreas) or for a metastatic niche.
Thank you very much for this suggestion.
Please, look at the answer to question number 8.
6. As the authors point out, CAPAN-1 cells migrate as a packed sheet, and this collective migration implies that cell-cell contact is maintained to a certain extent. As it is discussed, it could be an effect of partial-EMT, and more insights could be gained by looking at the expression of E-cadherin of cells in the sheets (for example by immunofluorescence).
We greatly appreciate this suggestion from the reviewer. However, to be able to carry it out, we would need more time than allotted to answer this question.
7. It would be interesting also to see if these findings would be validated by co-culturing tumor cells with pancreatic stellate cells.
As the reviewer has previously pointed out, the stroma is made up of a complex microenvironment integrated by many cell types. There is no doubt that one of the most important components are the stellate cells, although it has been shown that the majority cells are the fibroblasts. This is the reason why we decided to carry out our co-culture model. Repeating the experimentation with stellate cells would provide new information, but in the same way we could not overcome the obstacle of the complexity of the tumor microenvironment. Therefore, we indicate this in the limitations of our study, and following the reviewer's instructions we add stellate cells as one of the main components of the tumor microenvironment:
“Finally, an important limitation of our study is that it focuses on cancer-associated fibroblasts as the leading stromal element interacting with pancreatic tumor cells. However, many other cell types in the stroma are known to influence tumor cells behavior, such as immune and inflammatory cells, adipocytes, stellate cells, and endothelial cells”.
8. Finally, it is curious the different effect that the two tumor cell lines can have on fibroblasts migration. They are both classified as epithelial, but Capan1 are isolated from a metastatic site, while PL-45 are isolated from the primary tumor. This could be one reason of the different behavior, as they would interact with different microenvironments. It could be speculated that cells in the metastatic site attract fibroblasts to them, while cells from the primary tumor are more stimulated by the fibroblasts to start invasion. A broader panel of cell lines could offer the opportunity to find patterns to better understand the tumor-stromal interaction.
The authors greatly appreciate this reviewer's comment, mainly because it has given us an additional insight into our work that we had not previously contemplated. We consider it extremely important. Therefore we have added to the Discussion section the following statement:
“On the other hand, we must not forget the different origin of the tumor cells used in this study: PL-45 cells come from a primary tumor, while Capan-1 from a metastatic site. Given the results obtained in our study, we could speculate that cells in the metastatic site attract fibroblasts to them, while cells from the primary tumor are more stimulated by the fibroblasts to start invasion. Therefore, fibroblasts would have dissimilar migratory responses upon stimulation with different pancreatic cancer cells, such as Capan-1 or PL-45, and different microenvironments”.
We have also provided additional information about the tumor lines used in this work, with bibliography supporting these data (Table S1).
Table S1. Differential characteristics between tummor cell lines included in the study
|
Capan-1 |
PL-45 |
References |
Tissue |
Pancreas; derived from metastatic site (liver) |
Pancreatic adenocarcinoma (primary tumor) |
https://www.lgcstandards-atcc.org/products/all/HTB-79.aspx?geo_country=es# https://www.lgcstandards-atcc.org/products/all/CRL-2558.aspx?geo_country=es#characteristics |
K-RAS |
Mutation |
Mutation |
Berrozpe et al. 1994 Jaffee et al. 1998 Kita et al. 1999 Butz et al. 2003 Li et al. 2010 |
CDKN2A/p16 |
Homozygous deletion |
Promoter methylation |
Caldas et al. 1994 Huang et al. 1996 Li et al. 2010 |
TP53 |
Mutation |
Mutation |
Berrozpe et al. 1994 Huang et al. 1996 Li et al. 2010 |
SMAD4/DPC4 |
Null |
Wild type |
Schutte et al. 1996 Su et al. 2001 Deer et al. 2010 Li et al. 2010 Dempe et al. 2010 |
Berrozpe G, Schaeffer J, Peinado MA, Real FX, Perucho M. Comparative analysis of mutations in the p53 and K-ras genes in pancreatic cancer. Int J Cancer. 1994; 58(2):185-91.
Butz J. Wickstrom E, Edwards J. Characterization of mutations and loss of heterozygosity of p53 and K-ras2 in pancreatic cancer cell lines by immobilized polymerase chain reaction. BMC Biotechnology. 2003; .3-11.
Caldas C, Hahn SA, da Costa LT, Redston MS, Schutte M, Seymour AB, et al. Frequent somatic mutations and homozygous deletions of the p16 (MTS1) gene in pancreatic adenocarcinoma. Nat Genet. 1994; 8(1):27-32.
Deer EL, Gonzalez-Hernandez J. Coursen JD, Shea JE, Ngatia J, Scaife CL, et al. Phenotype and genotype of pancreatic cancer cell lines. Pancreas. 2010; 39(4):425-35.
Dempe S. Stroh-Dege AY, Schwarz E, Rommelaere J, Dinsart C. SMAD4: a predictive marker of PDAC cell permissiveness for oncolytic infection with parvovirus H-1PV. Int J Cancer. 2010; 126(12):2914-27.
Huang L, Goodrow TL, Zhang SY, Kein-Szanto AJ, Chang H, Ruggeri BA. Deletion and mutation analyses of the P16/MTS-1 tumor suppressor gene in human ductal pancreatic cancer reveals a higher frequency of abnormalities in tumor-derived cell lines than in primary ductal adenocarcinomas. Cancer Res. 1996; 56(1):1137-41.
Jaffee EM, Schutte M, Gossett J, Morsberger LA, Adler AJ, Thomas M, et al. Development and characterization of a cytokine-secreting pancreatic adenocarcinoma vaccine from primary tumors for use in clinical trials. Cancer J Sci Am. 1998; 4(3):194-203.
Kita K, Saito S, Morioka CY, Watanabe A. Growth inhibition of human pancreatic cancer cell lines by anti-sense oligonucleotides specific to mutates K-ras genes. Int J Cancer. 1999; 80(4):553-8.
Li J. Wientjes MG, Au JL. Pancreatic cancer: pathobiology, treatment options, and drug delivery. AAPS J. 2010; 12(2):223-32.
Schutte M. Hruban RH, Hedrick L, Cho KR, Nadasdy GM, Weinstein CL, et al. DPC4 gene in various tumor types. Cancer Res. 1996; 56(11):2527-30.
Su GH, Gansal R, Murphy KM, Montgomery E, Yeo CJ, Hruban RH, et al. ACVR1B (ALK4, activin receptor type 1B) gene mutations in pancreatic carcinoma. Proc Natl Acad Sci USA. 2001; 98(6):3254-7.
Specific points about the figures:
1. Figures 2A shows images of direct co-cultures of PDAC-fibroblast cells, but the 20:80 ratio does not appear evident (more likely 50:50). Please add cell number quantification in direct co-cultures reflecting 20:80 ratio as declared in the experimental design.
Thanks for this remark. In order to show fibroblasts activation, we chose images in which enough cells of both types could be seen. We have added Figure 2A, in which we verify our assertion that cultures were in a ratio of 20:80 (tumor cell: fibroblast).
2. In Figure 3B E-cadherin expression was detected via Western blot where the decrease in E-cadherin expression was indicated to be attributed to an EMT process. Was vimentin expression also determined before?
As we explained in answer 3 above, we performed a stripping of the membranes in which the expression of E-cadherin was verified. Afterwards, we hybridized them with an anti-Vimentin antibody. Unfortunately we were unable to see the expression of this protein in any of the conditions studied in these membranes. In our opinion, this was due to the fact that antibody was not working properly. We are aware of the importance of this result, but we would have to acquire a new antibody and perform the full experiment again. If this information were absolutely necessary, we would need more time to carry out the experimentation.
3. Figures 4E-G, and the relative paragraph 3.4, explaining the migratory capacity of cells are hard to understand and quantify. Please include graphs with percentages of migration for each cell line and condition to help the reader easily interpret the results.
Following your instructions, we have added to figure 4 the charts with percentages of migration of each cell line compared to their respective controls. Migration was quantified by counting the 10 cells furthest from the edge of the wound in 5 randomly fields from 3 different photographs. We have also clarified the text, indicating what each image corresponds to. We hope this will help to better understand such results.
4. Description of figure 4F: it is stated that the control in this experiment were tumor cells in growth medium. Since the experiment was investigating the effect of tumor cells on invasion of LC5-GFP+, a more appropriate control would be LC5-GFP+ in growth medium.
We thank the reviewer for pointing out this mistake in the text. Indeed, in this image the control in the experiment are fibroblast cells in growth medium. We have corrected the figure caption.
5. In Figure 5 A-E (TEM images) arrows pointing at extracellular vesicles (if visible) would help the reader understanding the figure.
Due to the size range (30-150 nm) of the extracellular vesicles, it is not possible to mark them on the photographs. Therefore, such a mark would be merely speculative. The images in Figure 5A-E are illustrative of the amount of vesicles that can be seen around cells. For extracellular vesicles isolation and quantification ExoQuick-TC® ULTRA EV Kit and ExoELISA-ULTRA CD63 Kit (System Biosciences, Palo Alto, CA) respectively were used.
Finally, we would like to thank the reviewer for all the help shown to improve our manuscript. We hope that our answers will be to your satisfaction.

Reviewer 2 Report
The authors aimed to study tumor-stromal interactions in an in vitro co-culture model between human PDAC cells (Capan-1 or PL-45) and fibroblasts (LC5).
Comments
- The authors should report why they selected Capan-1 or PL-45 for performing their work.
- The same should be reported for the stromal tissue.
- The same experiments should be carried out in different type of human PDAC
Author Response
- The authors should report why they selected Capan-1 or PL-45 for performing their work.
Following their instructions, we have added a supplementary table (Table S1) in which we indicate the differences between the Capan-1 and PL-45 cell lines and that motivated us to choose them to carry out this study.
Table S1. Differential characteristics between tummor cell lines included in the study
|
Capan-1 |
PL-45 |
References |
Tissue |
Pancreas; derived from metastatic site (liver) |
Pancreatic adenocarcinoma (primary tumor) |
https://www.lgcstandards-atcc.org/products/all/HTB-79.aspx?geo_country=es# https://www.lgcstandards-atcc.org/products/all/CRL-2558.aspx?geo_country=es#characteristics |
K-RAS |
Mutation |
Mutation |
Berrozpe et al. 1994 Jaffee et al. 1998 Kita et al. 1999 Butz et al. 2003 Li et al. 2010 |
CDKN2A/p16 |
Homozygous deletion |
Promoter methylation |
Caldas et al. 1994 Huang et al. 1996 Li et al. 2010 |
TP53 |
Mutation |
Mutation |
Berrozpe et al. 1994 Huang et al. 1996 Li et al. 2010 |
SMAD4/DPC4 |
Null |
Wild type |
Schutte et al. 1996 Su et al. 2001 Deer et al. 2010 Li et al. 2010 Dempe et al. 2010 |
Berrozpe G, Schaeffer J, Peinado MA, Real FX, Perucho M. Comparative analysis of mutations in the p53 and K-ras genes in pancreatic cancer. Int J Cancer. 1994; 58(2):185-91.
Butz J. Wickstrom E, Edwards J. Characterization of mutations and loss of heterozygosity of p53 and K-ras2 in pancreatic cancer cell lines by immobilized polymerase chain reaction. BMC Biotechnology. 2003; .3-11.
Caldas C, Hahn SA, da Costa LT, Redston MS, Schutte M, Seymour AB, et al. Frequent somatic mutations and homozygous deletions of the p16 (MTS1) gene in pancreatic adenocarcinoma. Nat Genet. 1994; 8(1):27-32.
Deer EL, Gonzalez-Hernandez J. Coursen JD, Shea JE, Ngatia J, Scaife CL, et al. Phenotype and genotype of pancreatic cancer cell lines. Pancreas. 2010; 39(4):425-35.
Dempe S. Stroh-Dege AY, Schwarz E, Rommelaere J, Dinsart C. SMAD4: a predictive marker of PDAC cell permissiveness for oncolytic infection with parvovirus H-1PV. Int J Cancer. 2010; 126(12):2914-27.
Huang L, Goodrow TL, Zhang SY, Kein-Szanto AJ, Chang H, Ruggeri BA. Deletion and mutation analyses of the P16/MTS-1 tumor suppressor gene in human ductal pancreatic cancer reveals a higher frequency of abnormalities in tumor-derived cell lines than in primary ductal adenocarcinomas. Cancer Res. 1996; 56(1):1137-41.
Jaffee EM, Schutte M, Gossett J, Morsberger LA, Adler AJ, Thomas M, et al. Development and characterization of a cytokine-secreting pancreatic adenocarcinoma vaccine from primary tumors for use in clinical trials. Cancer J Sci Am. 1998; 4(3):194-203.
Kita K, Saito S, Morioka CY, Watanabe A. Growth inhibition of human pancreatic cancer cell lines by anti-sense oligonucleotides specific to mutates K-ras genes. Int J Cancer. 1999; 80(4):553-8.
Li J. Wientjes MG, Au JL. Pancreatic cancer: pathobiology, treatment options, and drug delivery. AAPS J. 2010; 12(2):223-32.
Schutte M. Hruban RH, Hedrick L, Cho KR, Nadasdy GM, Weinstein CL, et al. DPC4 gene in various tumor types. Cancer Res. 1996; 56(11):2527-30.
Su GH, Gansal R, Murphy KM, Montgomery E, Yeo CJ, Hruban RH, et al. ACVR1B (ALK4, activin receptor type 1B) gene mutations in pancreatic carcinoma. Proc Natl Acad Sci USA. 2001; 98(6):3254-7.
- The same should be reported for the stromal tissue.
We chose fibroblasts as the majority cells within the cellular microenvironment. However, we are aware that this is not the only type of cells that make up the tumor stroma, therefore, in the Discussion Section, we indicate it as a limitation of our work.
“Finally, an important limitation of our study is that it focuses on cancer-associated fibroblasts as the leading stromal element interacting with pancreatic tumor cells. However, many other cell types in the stroma are known to influence tumor cells behavior, such as immune and inflammatory cells, adipocytes, stellate cells, and endothelial cells. On the other hand, fibroblasts used in this work are embryonic lung cells without prior contact with primary tumors or metastatic niches, which acquired CAF conditions in co-cultures with pancreatic tumor cells. Two different PDAC cell lines were used in our co-culture model, and we cannot generalize the results to the entire spectrum of pancreatic cancer. In this study, we have achieved some relevant results, but also accepting the incomplete approximation of the in vitro set-ups, as they are not always able to simulate the high tissue complexity found in vivo”.
- The same experiments should be carried out in different type of human PDAC.
We agree with the reviewer that the results would be stronger if they were carried out in a wide panel of pancreatic tumor lines with different molecular characteristics. However, this requirement is out of our scope since it entails doing all the experiments again, and it would take us much longer than accorded to respond to the reviewers.
Finally, we would like to thank the reviewer for all the help shown to improve our manuscript. We hope that our answers are to your satisfaction.

Round 2
Reviewer 1 Report
The authors replied to most of my questions and the article could be accepted for publication
Reviewer 2 Report
I have no further comments